# Prevalence and factors associated with psychological distress among key populations in Nigeria

Bartholomew Ochonye[1], Godwin Emmanuel[1], Roger Abang[1], Olaniyi Felix Sanni[1]*, Paul Umoh[1], Abiye Kalaiwo[2], Nanribet Mwoltu[1], Paul Amechi[1], Olugbemi Motilewa[3]

1 Research and Development Department, Heartland Alliance, Abuja, Nigeria, 2 Program Management, USAID, Abuja, Nigeria, 3 Department of Community Medicine, University of Uyo Teaching Hospital, Uyo, Akwa Ibom State, Nigeria

☯ These authors contributed equally to this work.

* sfelix@heartlandalliancenigeria.org

## Abstract

### Background

Stigmatization and discrimination within healthcare settings deter key populations (KPs) from seeking mental health and psychosocial support (MHPS). Consequently, understanding the prevalence, associated factors, and impact of the MHPSS intervention on psychological distress among Nigeria's KPs is crucial.

### Method

This is a cross-sectional study focused on KPs, including Female Sex Workers (FSW), Men who have Sex with Men (MSM), and People Who Inject Drugs (PWID) enrolled in Heartland Alliance LTD/GTE across 17 One-Stop Shops (OSS) in six states of Nigeria. Data were extracted from the databases of the OSS. PD was assessed using the Mental Health Screening Form III (MHSF-III). Descriptive statistics and univariable and multivariable binary logistic regression models were done using IBM-SPSS version 28.

### Results

The prevalence of PD among the KPs was 9.7%. Higher rates were observed among FSWs (12.0%). Of the 22310 KPs, the prevalence of PD was 9.7%. The major dependants of PD include being a PWID with PD prevalence of 8.5% and AOR of 1.95 (95% CI: 0.60–0.98, p = 0.015), alcohol intake with PD prevalence of 97.7% and AOR of 21.83 (95% CI: 15.13–56.83, p<0.001), and having experienced gender-based violence with PD prevalence of 99.0% and AOR of 25.70 (95% CI: 17.10–38.73, p<0.001). All Participants (100%) were given brief intervention, and 1595 of 2159 (73.8%) were referred for further psychological intervention. The services with the highest proportion were psychoeducation (21.20%), followed by coping skills training (17.70%) and motivational enhancement (12.90%).

**Data Availability Statement:** All relevant data are within the manuscript and its Supporting Information files.

**Funding:** Funding for this project was provided by the United States Agency for International Development (USAID) in the form of a contract agreement award (72062020CA00001) received by AK. The funding from the USAID is for the Key Populations Community HIV Services, Action, and Response (KPCARE 1) project implemented by Heartland Alliance Ltd/GTE. Data used for analysis were made available by Heartland Alliance LTD/GTE (HALG). No additional external funding was received for this study.

**Competing interests:** The authors have declared that no competing interests exist.

**Abbreviations:** FSW, - Female Sex Workers; KP, Key population; MHPS, Mental health and psychosocial support; MSM, Men who have Sex with Men; OSS, One Stop Shop; PD, Psychological Distress; PWID, People Who Inject Drugs.

## Conclusion

The study highlights the critical need for targeted mental health interventions among KPs in Nigeria, primarily focusing on those with a history of substance abuse and gender-based violence. Despite universal brief interventions, the proportion enrolled in further Psychosocial support indicates a need to improve mental health service utilization among the KPs in Nigeria.

## Introduction

Mental and Psychosocial well-being are fundamental aspects of human health, influencing overall quality of life, resilience, and the ability to thrive [1]. Among marginalized and vulnerable populations, such as female sex workers (FSW), men who have sex with men (MSM), people who inject drugs (PWID), and transgender individuals, the challenges to mental health are often exacerbated by stigma, discrimination, and marginalization [2].

Nigeria, a diverse and populous nation, is home to various key populations (KPs) facing distinct challenges related to their identities and activities. Sex workers often confront pervasive stigma, violence, and discrimination, contributing to mental health issues such as depression, anxiety, and post-traumatic stress disorder (PTSD) [3]. MSM encounter legal and social challenges, including criminalization and discrimination, resulting in significant mental distress and substance abuse [4]. In addition to battling addiction, PWID Transgender individuals grapple with high levels of discrimination, violence, and a lack of understanding, resulting in pronounced mental health disparities [5]. According to the American Psychological Association (APA), psychological distress is described as a collection of distressing mental and physical symptoms related to the regular mood fluctuations experienced by most people [6]. Individuals with psychological distress in these groups may encounter interruptions in numerous aspects of their lives, such as relationships, employment or school aspirations, and personal well-being [7].

With the One-Stop-Shops' (OSS) purview, mental health and Psychological support services (MHPSS) take on various forms designed to address the mental health challenges specific to KPs [8]. Counselling services are at the core of MHPSS, providing a safe, confidential space for individuals to discuss their mental health concerns and receive guidance and support from trained professionals. Group therapy sessions foster community and solidarity among KPs, facilitating open dialogue and reducing feelings of isolation [8]. Many OSS prioritize mental health education as a preventive measure to increase awareness of mental health issues, combat stigma, and promote self-care practices [9]. Additionally, these centres are crucial in connecting KPs with external mental health professionals and facilities when specialized interventions are required, ensuring comprehensive care is accessible [9].

Despite the noble objectives of OSS, several challenges hinder the effectiveness of MHPSS delivery. Stigmatization and discrimination within healthcare settings continue to deter KPs from seeking MHPSS, necessitating comprehensive training of healthcare providers in cultural competency and sensitivity [10]. Limited funding and resources pose significant barriers to the availability and quality of MHPSS [11], emphasizing the need for increased funding for mental health service's legal framework, includes provisions that criminalize the activities of KPs, creating a climate of fear and hindering access to healthcare services without fear of legal repercussions. Advocacy and legal reform are imperative to safeguard the rights of these individuals and ensure their access to healthcare. Furthermore, disparities in MHPSS availability persist

between urban and rural areas in Nigeria, necessitating efforts to establish equitable access to these services across all country regions. The study provides a complete overview of the prevalence, associated factors, and impact of the MHPSS intervention on psychological distress among Nigeria's Key Populations.

## Methodology

### Study design

This study takes a mixed-methods approach, combining elements of cross-sectional and quasi-experimental study designs to investigate the prevalence, associated factors, and impact of a Mental Health and Psychosocial Support (MHPSS) intervention on psychological distress (PD) among Key Populations in Nigeria. The study focused on various groups such as FSW, MSM, and PWID. These groups are all participants in the Heartland Alliance LTD/GTE One Stop Shops program, which operates in multiple states across Nigeria. A one-stop shop is a concentrated service centre that provides diverse services or products, making it a handy and efficient choice for customers or clients.

### Study population

The study used a retrospective approach, using data gathered from HA LTD/GTE OSS databases across Nigeria's designated states. A purposive sampling was employed to select only FSW, MSM, and PWID. The aim was to provide a thorough understanding of psychological distress prevalence and its associated factors among KPs. Data were extracted from all 17 OSS in six Nigerian states: Lagos, Bayelsa, Cross River, Akwa Ibom, Jigawa, and Niger. The data was assessed for analysis in August 2023.

### Inclusion and exclusion criteria

**Inclusion criteria.**   • KP's in the category of MSM, FSW and PWID

- Must be aged 18 and above.

- Must be accessing MHPSS service from HALG

**Exclusion criteria.**

- Other KP's such as Transgenders and PED.

**Definition of variables.**

- Sociodemographic factors such as age, gender, key population, state of residence, HIV status, alcohol and drug use, and experience of gender-based violence were defined to explore the factors influencing PD prevalence among KP's. Also, the association between the sociodemographic factors and PD was conducted by providing both Unadjusted Odd ratio (OR) and Adjusted Odd ratio (AOR) with a corresponding 95% Confidence interval (Cl). Additionally, variables such as session modality, the number of sessions attended and the current level of respondents' insight into their PD issues contributed to understanding the intervention's impact on individuals experiencing PD.

**Data collection.**   The assessment tool used by Heartland Alliance LTD/GTE for evaluating mental health was the Mental Health Screening Form III (MHSF-III). It was developed to

screen for possible mental health problems, particularly in substance abuse treatment [12]. This questionnaire consists of "18 "yes" and "no" questions related to the respondent's history. It has a maximum score of 18, with question 6 having two parts [12]. The MHSF-III is scored by totalling the "yes" responses (1 point each) for a maximum score of 17. It includes questions 3 through 17 [12]. If specific questions yield positive responses, seeking consultation with a mental health specialist for further evaluation is advisable. The form also provides a total score, facilitating research and evaluation of the relationship between mental health and substance dependence in the program. Research has shown that most MHSF-III items perform well for screening purposes, providing evidence of their validity [12]. The questionnaire was self-administered, allowing participants to independently reply to the "18" yes/no questions about their mental health history.

**Data analysis.** Data were cleaned before being imported to a spreadsheet and analyzed with IBM-SPSS version 28. To summarise the demographic and health metric data, descriptive statistics were used. Logistic regression analysis, incorporating both Odds Ratio (OR) and Adjusted Odds Ratio (AOR), was used to determine any association between demographics and PD prevalence, defined in this context as respondents enrolled for intervention after further psychological intervention. Chi-squared tests investigated any links between PD screening findings and treatment outcomes.

**Ethical consideration.** Heartland Alliance's authorized KP-CARE 1 ethical clearance issued by the Uyo Institutional Health Research Ethical Committee (IHREC) was used for project implementation research. Per HALG's authorization to access KP's data, this clearance was changed to maintain high data protection and security standards. To protect participant confidentiality and privacy, strict steps were implemented to anonymize and securely preserve the obtained data. Additionally, both written and verbal consent were obtained from the study participants. The consent form was written in English and translated into Pidgin English, a general language that all Nigerians understand. For those who could not read, the protocol was read to them until they understood, and those who agreed to participate in the study were made to thumbprint the consent form.

## Results

### Prevalence and significance of psychological distress (PD) by Sociodemographic of KP

The Sociodemographic and prevalence of PD among KPs across different states in Nigeria is illustrated in Table 1. The study included 22,310 Key Populations (KPs) who used HALG services between September 2022 and July 2023, a period of ten months. The average age of participants was 31 years, with a standard deviation of 7.3. The age distribution showed that the majority fell within the 21–30 years range (50.8%), followed by the 31–40 years range (34.4%). There were 48.5% males and 51.5% females among the total subjects, representing three (3) separate KP groups- MSM (37.1%), FSW (44.5%) and PWID (18.5%). The KP's spread across six (6) states, with cross rivers (36.2%) and Jigawa (12.7%) recording the highest population. The majority of Key Populations (KPs) were HIV positive (81.5%). However, alcohol and drug usage (0.6%) and gender-based violence (0.9%) among the participants were relatively low.

Of the 22310 KPs, the prevalence of PD was 9.7% (2159 KPs). PD was higher among females (12.1%) than males (7.1%). PD was highest among FSW (12.0%), followed by PWID (8.5) and MSM (7.5%) in the key population. Participants aged 41–50 (14.6%) exhibited the highest PD in this category. Akwa Ibom (53.6%) and Lagos state (31.6%) recorded the highest prevalence of PD compared to other states. PD was higher in respondents who tested positive for HIV (11.6%) than those who tested Negative (1.4%). PD was more prevalent among KPs with a

**Table 1. Sociodemographic characteristics and prevalence of PD among KPs enrolled in HA LTD/GTE OSS across Nigeria (September 2022 and July 2023).**

| Sociodemographic Variable | | Frequency (%) (n = 22310) | PD Prevalence (%) (n = 2159) | $X^2$ (P-value) |
|---|---|---|---|---|
| | Overall | 22310 | 2159 (9.7%) | |
| Age Category | ≤ 20 | 938 (4.2) | 119 (12.7) | 90.417 (<0.001*) |
| | 21–30 | 11325 (50.8) | 953 (8.4) | |
| | 31–40 | 7670 (34.4) | 747 (9.7) | |
| | 41–50 | 2073 (9.3) | 303 (14.6) | |
| | 50 + | 304 (1.4) | 37 (12.2) | |
| Gender | Male | 10813 (48.5) | 773 (7.1) | 153.471 (<0.001*) |
| | Female | 11497 (51.5) | 1386 (12.1) | |
| Key Population | MSM | 8273 (37.1) | 620 (7.5) | 110.607 (<0.001*) |
| | FSW | 9919 (44.5) | 1187 (12.0) | |
| | PWID | 4118 (18.5) | 352 (8.5) | |
| State | Bayelsa | 1939 (8.7) | 9 (0.5) | 6611.479 (<0.001*) |
| | Akwa Ibom | 1911 (8.6) | 1025 (53.6) | |
| | Cross River | 8069 (36.2) | 177 (2.2) | |
| | Jigawa | 2833 (12.7) | 0 (0.0) | |
| | Lagos | 2222 (10.0) | 702 (31.6) | |
| | Niger | 5336 (23.9) | 246 (4.6) | |
| HIV Status | Negative | 4129 (18.5) | 56 (1.4) | 401.355 (<0.001*) |
| | Positive | 18181(81.5) | 2103 (11.6) | |
| Alcohol and Drug use | No | 22180 (99.4%) | 2032 (9.2) | 1158.897 (<0.001*) |
| | Yes | 130 (0.6) | 127 (97.7) | |
| Gender-Based Violence | No | 22117 (99.1) | 1968 (8.9) | 1775.627 (<0.001*) |
| | Yes | 193 (0.9) | 191 (99.0) | |

* Statistical significance (P≤0.05)

history of alcohol and drug use (97.7%) as well as those with a history of gender-based violence (99.0%), P<0.001.

## Association between sociodemographic factors and psychological distress among KP's enrolled in OSS

Table 2 shows that female KPs had a higher prevalence at 12.1%. The OR was 1.78 (95% CI: 1.62–1.95, P<0.001). However, upon adjustment for other variables, the OR dropped to 0.82 (95% CI: 0.45–1.51) and was no longer statistically significant (p = 0.527). MSM had a 7.5% MPHD prevalence. FSW had a 12.0% prevalence and an OR of 1.68 (95% CI: 1.52–1.86, p<0.001). However, the adjusted odds ratio was 1.52 (95% CI: 0.76–3.05, p = 0.240), becoming non-significant. PWID had an 8.5% prevalence with an OR of 1.15 (95% CI: 1.01–1.32, p = 0.040). Interestingly, the AOR increased to 1.95 (95% CI: 0.60–0.98, p = 0.015), becoming statistically significant. When other variables are accounted for, age was not a robust predictor for PD.

Akwa Ibom showed a high prevalence of 53.6% with an OR of 24.09 (95% CI: 12.10–48.47, p<0.001) and an AOR of 27.09 (95% CI: 17.06–67.84, p<0.001), both highly significant than Bayelsa. HIV- positive KPs had an 11.6% prevalence of PD. The OR was significantly high at 9.513 (95% CI: 7.280–12.432, p<0.001). However, after adjustment, the OR plummeted to 0.74 (95% CI: 0.36–1.55) and was no longer statistically significant.

**Table 2. Association between sociodemographic factors and PD among KP's enrolled in OSS.**

| Variable | PD | Unadjusted | | Adjusted | |
|---|---|---|---|---|---|
| | Prevalence (%) | OR (95% CL) | P value | OR (95% CL) | P value |
| **Gender** | | | | | |
| Male | 773 (7.1) | Ref | - | Ref | - |
| Female | 1386 (12.1) | 1.78 (1.62–1.95) | <0.001* | 0.82 (0.45–1.51) | 0.527 |
| **Key Population** | | | | | |
| MSM | 620 (7.5) | Ref | - | Ref | - |
| FSW | 1187 (12.0) | 1.68 (1.52–1.86) | <0.001* | 1.52 (0.76–3. 05) | 0.240 |
| PWID | 352 (8.5) | 1.15 (1.01–1.32) | 0.040* | 1.95 (0.60–0.98) | 0.015* |
| **Age Category** | | | | | |
| ≤ 20 | 119 (12.7) | 1.05 (0.71–1.56) | 0.814 | 4.64 (1.05–20.64) | 0.44 |
| 21–30 | 953 (8.4) | 0.66 (0.47–0.94) | 0.021* | 2.87 (0.72–11.44) | 0.136 |
| 31–40 | 747 (9.7) | 0.78 (0.55–1.11) | 0.164 | 2.22 (0.56–8.86) | 0.259 |
| 41–50 | 303 (14.6) | 1.24 (0.86–1.78) | 0.256 | 3.08 (0.75–12.67) | 0.119 |
| 50 + | 37 (12.2) | Ref | - | - | - |
| **State** | | | | | |
| Bayelsa | 9 (0.5) | Ref | - | Ref | - |
| Akwa Ibom | 1025 (53.6) | 24.09 (12.10–48.47) | <0.001* | 27.09 (17.06–67.84) | <0.001* |
| Cross River | 177 (2.2) | 4.81 (128.10–480.47) | <0.001* | 13.94 (9.82–27.58) | <0.001* |
| Jigawa | 0 (0.0) | - | - | - | - |
| Lagos | 702 (31.6) | 29.04 (14.14–91.80) | <0.001* | 39.89 (26.36–93.34) | <0.001* |
| Niger | 246 (4.6) | 10.36 (5.32–20.19) | <0.001* | 13.08 (11.6–326.95) | <0.001* |
| **HIV Status** | | | | | |
| Negative | 56 (1.4) | Ref | - | Ref | - |
| Positive | 2103 (11.6) | 9.513 (7.280–12.432) | <0.001* | 0.74 (0.36–1.55) | <0.001* |
| **Alcohol and Drug use** | | | | | |
| No | 2032 (9.2) | Ref | - | Ref | - |
| Yes | 127 (97.7) | 19.75 (13.47–32.07) | <0.001* | 21.83 (15.13–56.83) | <0.001* |
| **Gender-Based Violence** | | | | | |
| No | 1968 (8.9) | Ref | - | Ref | - |
| Yes | 191 (99.0) | 17.759 (14.59–39.93) | <0.001* | 25.70 (17.10–38.73) | <0.001* |

* Statistical significance (p 0.05)

Among KPs taking alcohol, PD prevalence was 97.7% with an OR of 19.75 (95% CI: 13.47–32.07, p<0.001) and an adjusted odds ratio of 21.83 (95% CI: 15.13–56.83, p<0.001), both being statistically significant. Lastly, in the case of gender-based violence, the prevalence was nearly universal at 99.0%. The OR was 17.759 (95% CI: 14.59–39.93, p<0.001), and the adjusted OR was even higher at 25.70 (95% CI: 17.10–38.73, p<0.001), both indicating strong statistical significance.

## KP's enrolled for MHPSS intervention

All Participants (100%) were given brief intervention, and 1595 of 2159 (73.8%) were referred for further psychological intervention after the brief intervention. Fig 1 shows that 752 (47.1%) of 1595 were referred for further psychological intervention and eventually enrolled for interventions.

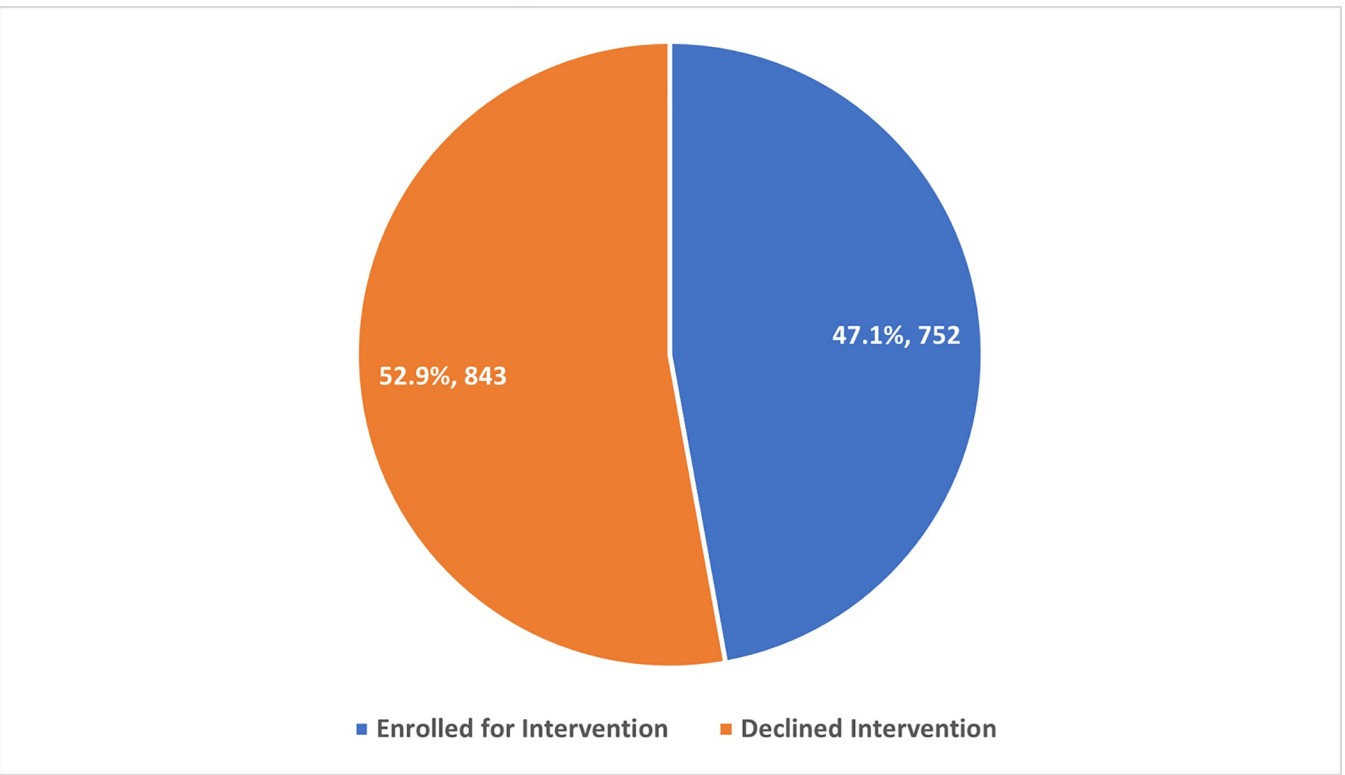

**Fig 1. KP's enrolled for MHPSS intervention.**

## Types of intervention received by KP with PD

Fig 2 illustrates different types of mental and psychological intervention services registered by participants in the intervention program. The services with the highest proportion were psychoeducation (21.20%), followed by coping skills training (17.70%) and motivational enhancement (12.90%).

## Characteristics of KPs with PD enrolled for interventions

The characteristics of Participants with PD were assessed in Table 3. The in-person session was conducted for 72.7% of those enrolled in the intervention; 24.6% were counselled virtually, and 2.7% in groups in the community. The majority of the participants attended only one session (74.0%) in comparison to those who added two sessions (25.5%) and three sessions (0.5%), respectively. More than half of the respondents (68.3%) had good insight, 23.0% had a fair level, and 8.6% lacked insights into their PD issues.

## Discussion

This study examines the prevalence of Mental health and psychological well-being issues and the types of interventions administered among KPs with PD in Nigeria. Mental health is increasingly recognized as a critical component of individual health. According to the World Health Organisation [13], mental health is an essential component of health, a state of physical, mental, social, and spiritual well-being and not merely the absence of disease or infirmity. On this premise, the government must promote, protect, and restore mental health since it may have significant consequences for the nation's progress due to disease, disability, and death

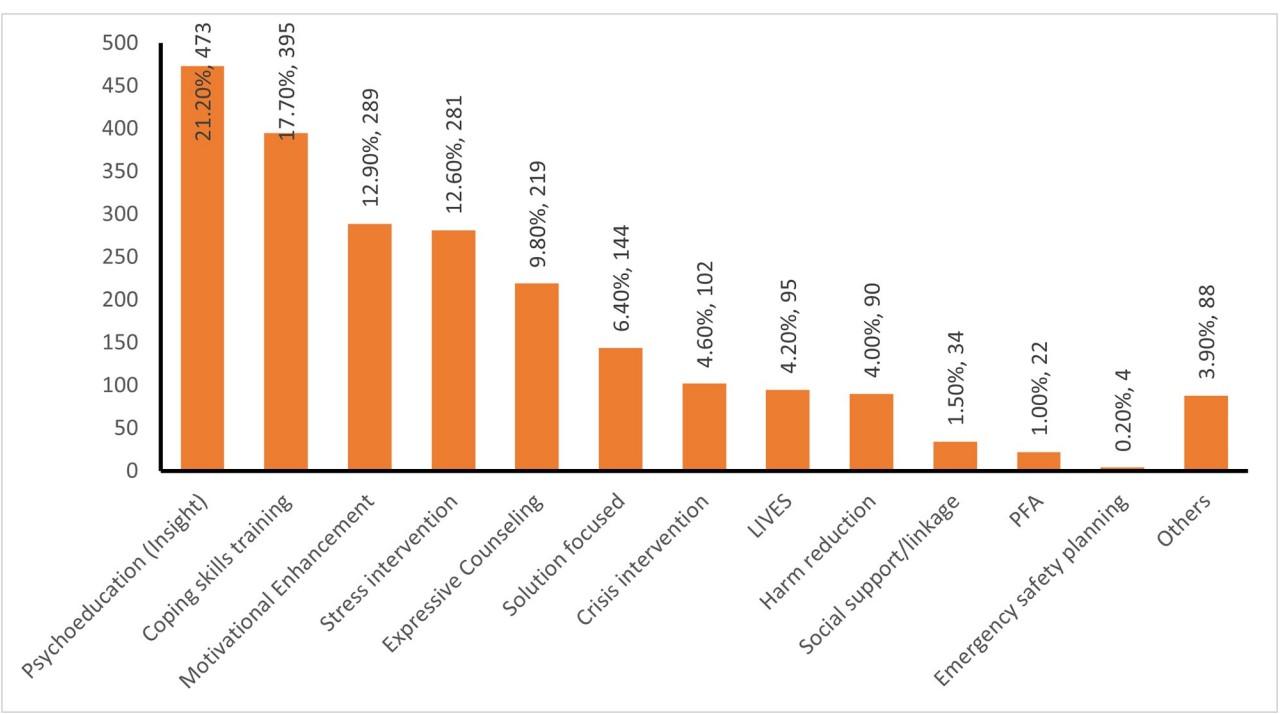

**Fig 2. Types of intervention received by KP with PD.**

among its people [14]. Nigerian society has numerous unfavourable attitudes and false notions about the mentally sick. Stigma is a huge issue in the country, causing people to be hesitant to seek treatment because they are embarrassed and fearful of prejudice and discrimination [14].

The study found varying levels of psychological distress among KPs in different states of Nigeria. This variation can be attributed to several factors, including sociodemographic characteristics and regional disparities in access to mental health services. Higher levels of psychological distress suggest a pressing need for targeted mental health interventions for KPs. A study by [15] showed that social determinants, including gender and region, often influence Psychological distress. Psychological distress was more prevalent among females (12.1%) than males (7.1%) among KPs. This gender disparity could be due to unique challenges and vulnerabilities female KPs face, such as sex work-related stigma and violence. Addressing the mental

**Table 3. Characteristics of KPs with PD enrolled for interventions.**

| Parameter | Variable | Enrolled KP's with Psychological distress % |
|---|---|---|
| | | (n = 752) |
| **Session Modality** | Group/community | 20 (2.7) |
| | In-person | 547 (72.7) |
| | Virtual | 185 (24.6) |
| **Number of sessions** | 1 | 556 (74.0) |
| | 2 | 192 (25.5) |
| | 3 | 4 (0.5) |
| **Current Level of Insight to PD issues** | Fair | 173 (23.0) |
| | Good | 514 (68.3) |
| | Lacking | 65 (8.6) |

health needs of female KPs should be a priority in mental health programs. A study by Kanayama et al. [16] found that female sex workers are at a higher risk of mental health issues due to gender-based violence and discrimination.

Among KPs, FSW had the highest prevalence of psychological distress (12.0%), followed by PWID (8.5%) and MSM (7.5%). Different KP groups may face distinct stressors, such as the stigma associated with sex work for FSW or drug-related challenges for PWID. Tailored mental health support programs should consider the specific needs of each KP subgroup. Research by Shareef [17] highlights the importance of understanding different groups' unique challenges in addressing mental health issues. In addition, participants aged 41–50 had the highest prevalence of psychological distress (14.6%) compared to other age groups. This could be linked to cumulative stressors over time or age-related health concerns. Mental health interventions for KPs should consider age-related factors in their approach. Studies by Carpenter et al. [18] suggest that older individuals in key populations may have distinct mental health needs due to their life experiences and health challenges.

Furthermore, Akwa Ibom (53.6%) and Lagos state (31.6%) had the highest prevalence of psychological distress among KPs. Regional differences may be attributed to variations in access to healthcare, cultural factors, and socioeconomic disparities. Identifying regions with higher prevalence can guide resource allocation for mental health services. A study by Adesina et al. [19] highlighted regional disparities in mental health outcomes in Nigeria. Also, participants who tested positive for HIV, had a history of substance use, or experienced gender-based violence had higher levels of psychological distress. Co-occurring health and psychological challenges can exacerbate psychological distress among KPs. Integrating mental health support into HIV care, substance use treatment, and gender-based violence prevention programs is crucial. Research by Azrin [20] emphasizes the need for integrated care for persons with multiple health and psychological concerns.

The study found that female participants had a higher chance of experiencing psychological distress than males. This finding aligns with existing literature that highlights the vulnerability of females to mental health issues due to various sociocultural and biological factors [21, 22]. It underscores the importance of gender-sensitive mental health interventions for female KPs within OSS programs. Further analysis revealed that among the KPs, FSWs were 1.519 times more likely, and PWID were 1.953 times more likely to experience psychological distress than MSM. This finding suggests that specific subpopulations within KPs may have distinct mental health needs and risk factors. Similar studies have demonstrated the unique challenges FSW and PWID face regarding mental health [23, 24]. Tailoring mental health support to these subgroups is imperative.

Participants aged 20 and below were found to be 4.644 times more likely to experience psychological distress than those aged 50 and above. This result highlights the vulnerability of younger individuals within KPs, which may be attributed to a lack of coping skills, social support, and stigma-related stress [25]. Mental health interventions targeting younger KPs are essential. The study revealed significant regional disparities, with KPs in Akwa Ibom and Lagos states facing significantly elevated odds of experiencing psychological distress compared to those in Bayelsa ($P < 0.001$). This geographic variation suggests that local contextual factors, healthcare infrastructure, and social support networks may contribute to mental health disparities [26]. Tailored interventions based on regional needs are warranted.

Additionally, participants with a positive HIV status were found to have a higher likelihood of experiencing psychological distress than those without the virus. This counterintuitive result may be attributed to the stigma and Psychological stressors associated with HIV diagnosis [27]. Comprehensive HIV care should incorporate mental health support to address this dual burden. Also, participants who engaged in the abuse of alcohol and drug substances were

more likely to experience psychological distress compared to those who did not. Additionally, participants with a history of Gender-Based Violence (GBV) were more likely to exhibit psychological distress. These findings underscore the interplay between substance abuse, GBV, and mental health issues among KPs [28]. Integrated interventions addressing both substance abuse and GBV alongside mental health are essential.

The findings indicate that all (100%) participants received a brief intervention, with 73.8% subsequently referred for further psychological intervention. Among those referred, 47.1% were eventually enrolled for comprehensive interventions. The reasons for the observed prevalence of PD enrolment among KPs in OSS are multifaceted. Understanding these factors is essential for tailoring interventions to address MHPSS needs effectively. One key factor contributing to the observed MHPSS enrolment rates may be destigmatization efforts integrated into One-Stop Shop services. Prior studies have emphasized the importance of creating safe, non-judgmental environments for people seeking help in mental health crises. The destigmatization efforts might have encouraged more participants to accept referrals for further psychological intervention. Another contributing factor could be the customization of interventions to meet the unique needs of KPs. Research has shown that tailored interventions are more effective in engaging and retaining marginalized populations in mental health services [29]. OSS may have successfully employed such approaches. Peer support programs within OSS may have played a significant role in encouraging enrolment for MHPSS interventions. Peer support has reduced barriers to mental health service utilization among KPs [30, 31].

The findings indicate that KPs enrolled in the One-Stop Shop program received a range of MHPSS interventions. The most prevalent interventions reported by participants were psychoeducation (21.2%), Coping skills training (17.7%), and Motivational enhancement (12.9%). The high prevalence of psychoeducation can be attributed to the effectiveness of this intervention in enhancing awareness and knowledge about mental health issues among KPs. Psychoeducation helps individuals understand their emotions, reduce stigma, and make informed decisions about their mental health. Previous research by Onnela et al. [32] found that psychoeducation significantly improved mental health awareness, leading to increased utilization of mental health services. In addition, coping skills training is essential for KPs who often face discrimination, stigma, and other stressors. This intervention equips them with practical strategies to manage stress, anxiety, and other psychological challenges that may arise due to their marginalized status. A study by Zakiyah and Rofi'ah [33] demonstrated that coping skills training effectively reduced stress and improved mental well-being among marginalized populations. The findings emphasize the importance of addressing stigma related to mental health among KPs. Increased psychoeducation can help reduce stigma and promote a more supportive environment for mental health services. Also, motivational enhancement is critical in engaging KPs in the MHPSS program and motivating them to participate actively. This finding suggests that KPs may require additional motivation and support to access mental health services. Salize et al. [34] conducted a study highlighting the importance of motivational enhancement in engaging hard-to-reach populations, such as KPs, in mental health programs.

The study found that most participants (72.7%) received in-person counselling, indicating a preference for face-to-face interaction. A smaller proportion (24.6%) received virtual counselling, while a minority (2.7%) participated in group sessions within the community. The preference for in-person counselling may be driven by the need for a safe and confidential space to discuss sensitive issues. Virtual counselling may face limitations in terms of privacy and technology access. This distribution may reflect participants' comfort levels with different counselling modalities and access to technology [35]. Also, the study revealed that 74.0% of participants attended only one counselling session.

In contrast, 25.5% attended two sessions, and a minimal 0.5% attended three. The high percentage of participants attending only one session could be due to the perceived effectiveness of a single session or practical constraints, such as work or other commitments. This distribution suggests that most KPs may perceive one session as sufficient for addressing their MHPSS needs, highlighting potential time and resource constraints. In contrast, this pattern of low utilization may be attributed to various factors, such as time constraints, stigma, and fear of disclosure.

Approximately 68.3% of participants demonstrated good insight into their PD issues, 23.0% exhibited fair insight, and 8.6% lacked insight. Training and awareness programs may influence disparity in insight levels to reduce stigma and increase mental health literacy among KPs. This variation in insight levels may be attributed to factors such as education, awareness, and stigma surrounding mental health, affecting individuals' ability to recognize and acknowledge their concerns. The study found that 86.6% of enrolled KPs experienced mild PD issues, 12.6% of individuals' issues, and only 0.5% reported severe issues. The predominance of mild symptoms may indicate that KPs seek help early, and One-Stop Shop interventions effectively address initial concerns before they escalate. The predominance of mild symptoms suggests that One-Stop Shop interventions may effectively address early-stage PD concerns among KPs. Service providers must assess and address this gap through targeted interventions and psychoeducation [36].

## Limitations of the study

One limitation of this study is the reliance on clinical data, which may not capture all relevant variables or nuances related to MHPSS among KPs. Since the study used existing data from Heartland Alliance LTD/GTE, it may not have been collected initially to address the specific research questions of this study, potentially limiting the depth of insights and the ability to explore certain factors in greater detail. Additionally, the study's focus on six selected Nigerian states may not fully represent the diversity of PD experiences and needs across the entire country, limiting the generalizability of the findings to the broader Nigerian population.

## Conclusion

The study provides valuable insights into the prevalence and factors associated with mental health and psychological support service utilization among KPs enrolled in One-Stop Shop interventions in Nigeria. The findings underscore the critical importance of addressing mental health within the broader context of individual well-being, aligning with the World Health Organization's perspective. Stigma remains a significant barrier to seeking mental health services in Nigeria, emphasizing the need for destigmatization efforts. The study reveals varying levels of psychological distress among KPs, influenced by sociodemographic factors, regional disparities, and unique challenges faced by specific subgroups. These findings highlight the necessity of tailoring mental health interventions to the specific needs of different KP subpopulations and age groups.

Furthermore, the study demonstrates the success of the One-Stop Shop program in engaging KPs in MHPSS interventions, with a preference for in-person counselling and a majority attending only one session. The predominance of mild symptoms suggests early intervention effectiveness. However, factors contributing to low session attendance warrant further investigation. Overall, this study underscores the urgency of addressing mental health issues among KPs in Nigeria and highlights the importance of tailored, destigmatized, and region-specific interventions to enhance the well-being of this vulnerable population. Future efforts should

focus on reducing barriers to access, increasing awareness, and integrating mental health support into existing KP programs.

## Supporting information

**S1 Data.**
(XLSB)

## Acknowledgments

A sincere gratitude to United States Agency for International Development (USAID) for the invaluable support in facilitating and contributing to this research. Their assistance has been instrumental in advancing this study.

## Author Contributions

**Conceptualization:** Bartholomew Ochonye, Godwin Emmanuel, Roger Abang, Olaniyi Felix Sanni.

**Data curation:** Godwin Emmanuel, Olaniyi Felix Sanni.

**Formal analysis:** Roger Abang, Olaniyi Felix Sanni.

**Investigation:** Godwin Emmanuel, Roger Abang, Nanribet Mwoltu, Olugbemi Motilewa.

**Methodology:** Paul Umoh, Abiye Kalaiwo.

**Software:** Olaniyi Felix Sanni.

**Supervision:** Roger Abang, Olaniyi Felix Sanni, Abiye Kalaiwo.

**Validation:** Godwin Emmanuel, Olaniyi Felix Sanni, Paul Amechi.

**Visualization:** Olaniyi Felix Sanni.

**Writing – original draft:** Roger Abang, Nanribet Mwoltu, Paul Amechi.

**Writing – review & editing:** Paul Umoh, Abiye Kalaiwo, Nanribet Mwoltu, Paul Amechi.

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
