## [Decision Letter · Decision Letter 0]

28 Dec 2023

PONE-D-23-38088Prevalence and Factors Associated with Psychological Distress among Key Populations in NigeriaPLOS ONE

Dear Dr. Sanni,

Thank you for submitting your manuscript to PLOS ONE. After careful consideration, we feel that it has merit but does not fully meet PLOS ONE’s publication criteria as it currently stands. Therefore, we invite you to submit a revised version of the manuscript that addresses the points raised during the review process.

The study objectives are critical toward improving pychosocial wellbeing of KPs as preventive measures that will promote good health and prevent spread of HIV and other related conditions. However, the paper requires major work to ensure these good objectives are clearly presented so that readers can make best use of the findings and ensure reproducibility. Therefore, please review carefully my comments below in addition to the 2 reviewers' comments.

We look forward to receiving your revised manuscript.

Kind regards,

Ibrahim Jahun, MD, MSC, PhD

Academic Editor

PLOS ONE

Journal Requirements:

3. Thank you for stating the following financial disclosure: "Funding and data used for analysis was made available by Heartland alliance LTD/GTE (HALG)".

4. Thank you for stating the following in the Acknowledgments Section of your manuscript: "A sincere gratitude to Heartland alliance LTD/GTE (HALG) for their invaluable support in facilitating and contributing to this research, their assistance has been instrumental in advancing this study."

Please remove any funding-related text from the manuscript and let us know how you would like to update your Funding Statement. Currently, your Funding Statement reads as follows: "Funding and data used for analysis was made available by Heartland alliance LTD/GTE (HALG)".

**Funding/Financial disclosure:**

- Please provide details of the funding source, Initials of authors who received each award, Full names of commercial companies that funded the study or authors, Initials of authors who received salary or other funding from commercial companies, URLs to sponsors’ websites.

-Also state whether any sponsors or funders (other than the named authors) played any role in: Study design, Data collection and analysis, Decision to publish, Preparation of the manuscript.

If they had no role in the research, include this sentence: “The funders had no role in study design, data collection and analysis, decision to publish, or preparation of the manuscript.”

If the study was unfunded, include this sentence as the Financial Disclosure statement: “The author(s) received no specific funding for this work."

- Please ensure tables, figures and outline including references are based on PLOS-ONE publication style which can be accessed through this link https://journals.plos.org/plosone/s/figures.

Additional Editor Comments:

**Technical Review:**

Introduction: will be good to provide how metal health challenges impact on KPs overall lifestyle and health outcomes and how findings from this study can help in improving their health.

Methods:

- Participants – rephrase the sentences to read like “methods” and not like “results”.

- Study design – rewrite this section by including details about the reason for purposive sampling, sampling technique, sample size e.t.c.

- Settings – Define what is one-stop-shop, objectives and modus operandi (briefly); provide justifications why the 6 states were selected. Did the 17 OSS comprise all OSS supported by this organization in the 6 states or there are other ones that have not been included? If yes, what are the inclusion/exclusion criteria in selecting the OSS?

- Data collection: some components of the study design where the questionnaire was described should be move to this section. Clearly describe how the data was collected – is it using CAPI, self-administered questionnaire e.t.c? The language used in administering the questionnaire, reverse translation (where applicable) e.t.c. This section can be merged with data analysis as Data Collection & Analysis. Also clearly outline variables used in each of the 2 analysis groups (descriptive and association). Define what is PD in the context of analyzed variables.

Results:

- Tables title/captions – indicate time period.

- Table 1 should indicate socio-demographic characteristics of the participants. Socio-economic factors are critical in PD and must be well analyzed here so that when describing study limitations’ case could be made about potential “confounders”.

- Also, variables like level of education, occupation are critical.

- The results section doesn’t flow well with the methods and introduction or the title of the study. Several other things including interventions are included in the results while they were not mentioned as part of the study objectives. The tile should clearly give the reader an idea of what to expect when reading the paper.

Discussions:

- The discussions require integration. Findings were discussed in “silos” under different subsections. The findings should be linked and integrated and discussed generally as a single study.

Reviewers' comments:

Reviewer's Responses to Questions

**Comments to the Author**

1. Is the manuscript technically sound, and do the data support the conclusions?

Reviewer #1: Yes

Reviewer #2: Partly

2. Has the statistical analysis been performed appropriately and rigorously? 

Reviewer #1: Yes

Reviewer #2: Yes

3. Have the authors made all data underlying the findings in their manuscript fully available?

Reviewer #1: Yes

Reviewer #2: No

4. Is the manuscript presented in an intelligible fashion and written in standard English?

Reviewer #1: Yes

Reviewer #2: Yes

5. Review Comments to the Author

Reviewer #1: The manuscript was well written with both descriptive and inferential statistical analysis done (Chi Squared test and Logistic regression and proper interpretation of the results).

However, The data analysis section of the methodology needs to be updated as this statement "Logistic regression analysis was used to determine any correlations between demographics and PD prevalence." is misleading and confusing. The use of the word "association" instead of "correlation" is more appropriate.

Also, the statement "The impact of health measures on PD results was investigated using multiple regression analysis" is too vague. Please provide more information and context to what the heath measures are.

Write OR and AOR in full on their first appearances in the manuscript with the acronym in the parenthesis.

Don't assume everybody knows their meaning.

Reviewer #2: Abstracts

• There was no information about sample size and the type of statistical analysis or test conducted.

• Arrange key words in alphabetical order of the first letters and confirm they are available on the MeSH database.

• The abstract needs to be rewritten to pass across the message in the full manuscript.

Introduction

• There should be a brief description of One-Stop-Shops' (OSS) as there are many references.

Methodology

• A methodology section should describe how you collected, organized, and analyse the data. The flow mostly starts with Study design, study population, inclusion and exclusion criteria, sampling methods, definition of variables, statistical analysis etc

• A result was wrongly presented in the methodology section, it should be reported in the appropriate research language at the result section.

• This language should be modified as there is no such thing as HALG services. There could be PSS or MHPSS services from HALG SDP

• This manuscript come across more like a quasi-experimental study than a cross sectional study, the author is advised to decide what the purpose of this study is.

Results

• The title of table 1 should be updated to reflect socio demographic characteristics.

• Why not start the table with socio demographic variables in this order Age group, Sex, Key Population type, State, etc

• What has description of intervention got to do with a cross sectional study?

Discussion

• The author should be consistent with the in-text citation style.

6. PLOS authors have the option to publish the peer review history of their article (what does this mean?). If published, this will include your full peer review and any attached files.

Reviewer #1: No

Reviewer #2: No

---

## [Author Response · Author response to Decision Letter 0]

2 Feb 2024

Revision done, a rebuttal letter have been submited

---

## [Decision Letter · Decision Letter 1]

6 Mar 2024

Prevalence and Factors Associated with Psychological Distress among Key Populations in Nigeria

PONE-D-23-38088R1

Dear Dr. Sanni,

Thank you for your patience. The two reviewers who reviewed your manuscript apparently were not able to locate your edited version (revised version) of the manuscript because it was placed at the very end of your PDF file. Therefore, the reviewers thought that you haven't addressed their comments. However, I found your edits and guided the reviewers on how to access it. Therefore, both reviewers have sent me mails confirming that you have fully addressed their comments. Please next time endeavor to arrange your submission in correct order to avoid confusion which may eventually lead to delays in making final decision. Therefore please disregard the comments from the reviewers below.

We’re pleased to inform you that your manuscript has been judged scientifically suitable for publication and will be formally accepted for publication once it meets all outstanding technical requirements.

Kind regards,

Ibrahim Jahun, MD, MSC, PhD

Academic Editor

PLOS ONE

Additional Editor Comments (optional):

Reviewers' comments:

Reviewer's Responses to Questions

**Comments to the Author**

1. If the authors have adequately addressed your comments raised in a previous round of review and you feel that this manuscript is now acceptable for publication, you may indicate that here to bypass the “Comments to the Author” section, enter your conflict of interest statement in the “Confidential to Editor” section, and submit your "Accept" recommendation.

Reviewer #1: (No Response)

Reviewer #2: (No Response)

2. Is the manuscript technically sound, and do the data support the conclusions?

Reviewer #1: Yes

Reviewer #2: Partly

3. Has the statistical analysis been performed appropriately and rigorously? 

Reviewer #1: Yes

Reviewer #2: (No Response)

4. Have the authors made all data underlying the findings in their manuscript fully available?

Reviewer #1: Yes

Reviewer #2: No

5. Is the manuscript presented in an intelligible fashion and written in standard English?

Reviewer #1: Yes

Reviewer #2: Yes

6. Review Comments to the Author

Reviewer #1: Most of the comments from the previous review was not addressed.

The authors should go through the feedback provided for the previous review and address the the comments.

Reviewer #2: All my earlier review comments below were not addressed by the author.

Abstracts

• There was no information about sample size and the type of statistical analysis or test conducted.

• Arrange key words in alphabetical order of the first letters and confirm they are available on the MeSH database.

• The abstract needs to be rewritten to pass across the message in the full manuscript.

Introduction

• There should be a brief description of One-Stop-Shops' (OSS) as there are many references.

Methodology

• A methodology section should describe how you collected, organized, and analyse the data. The flow mostly starts with Study design, study population, inclusion and exclusion criteria, sampling methods, definition of variables, statistical analysis etc

• A result was wrongly presented in the methodology section, it should be reported in the appropriate research language at the result section.

• This language should be modified as there is no such thing as HALG services. There could be PSS or MHPSS services from HALG SDP

• This manuscript come across more like a quasi-experimental study than a cross sectional study, the author is advised to decide what the purpose of this study is.

Results

• The title of table 1 should be updated to reflect socio demographic characteristics.

• Why not start the table with socio demographic variables in this order Age group, Sex, Key Population type, State, etc

• What has description of intervention got to do with a cross sectional study?

Discussion

• The author should be consistent with the in-text citation style.

7. PLOS authors have the option to publish the peer review history of their article (what does this mean?). If published, this will include your full peer review and any attached files.

Reviewer #1: No

Reviewer #2: No

---

## [Editor Report · Acceptance letter]

3 Apr 2024

PONE-D-23-38088R1 

PLOS ONE

Dear Dr. Sanni, 

I'm pleased to inform you that your manuscript has been deemed suitable for publication in PLOS ONE. Congratulations! Your manuscript is now being handed over to our production team.

Kind regards, 

on behalf of

Dr. Ibrahim Jahun 

Academic Editor

PLOS ONE